# Validation of a Simplified Method for Estimating the Harmonic Response of Rogowski Coils with the Monte Carlo Method

**DOI:** 10.3390/s24061746

**Published:** 2024-03-07

**Authors:** Christian Betti, Alessandro Mingotti, Roberto Tinarelli, Lorenzo Peretto

**Affiliations:** Department of Electrical, Electronic and Information Engineering, Guglielmo Marconi Alma Mater Studiorum, University of Bologna, Viale del Risorgimento 2, 40136 Bologna, Italy; christian.betti2@unibo.it (C.B.); alessandro.mingotti2@unibo.it (A.M.); lorenzo.peretto@unibo.it (L.P.)

**Keywords:** characterization, harmonics, Rogowski coil, simplified method, uncertainty evaluation

## Abstract

The need to monitor the power network is leading to a significant increase in the number of measurement points. These points consist of intelligent electronic devices and instrument transformers (or more in general sensors). However, as the number of devices increases, so does the demand for their characterization and testing. To this end, the authors formulated a new characterization procedure that offers numerous benefits for manufacturers and system operators. These benefits include: (i) reducing testing time (thus lowering costs), (ii) simplifying the existing procedures, and (iii) increasing the number of tested devices. In this study, to complete the validation of the proposed characterization procedure, the authors performed a comprehensive uncertainty evaluation. This included the identification and analysis of the uncertainty sources, the implementation of the Monte Carlo method to obtain the statistical parameters of the quantities of interest, and the final method assessment according to the obtained results. Each step is described in detail, and the results allow one to (i) replicate the uncertainty analysis on other types of instrument transformers and (ii) implement the proposed harmonic characterization procedure with the confidence that the method is accurate, flexible, and scalable.

## 1. Introduction

The environmental impact of power grids is influenced by various factors that make solving the problem intricate [1]. Initiatives aimed at reducing the repercussions on the environment are being implemented, such as the transition to renewable energy sources and energy efficiency, in order to mitigate the undesired effects of the production and transmission of electricity. However, the concerns of climate change must coexist with the increasing need to monitor the network, as ensuring its reliability and stability is crucial [2].

The power grid consists of numerous interconnected components, including power plants, transformers, transmission lines, and distribution networks. These components must all operate together seamlessly to provide a steady supply of electricity to customers. However, any interruptions or failures within the system can lead to power outages, which can have serious consequences for society, the economy, and public safety [3].

To prevent these issues, power grid operators adopt advanced monitoring systems to examine the grid performance in real time [4,5]. These systems collect data from sensors and other devices installed throughout the grid and employ sophisticated analytics to detect any anomalies or potential problems [6,7]. Therefore, thanks to early identification, grid operators can take corrective actions to avert power outages and ensure the reliability and stability of the grid.

Instrument transformers (ITs) are key elements of the power network, as they enable the measurement of voltages, currents, and consequently power. The importance of ITs has led researchers to produce a vast amount of literature on them. For example, in [8,9] the authors described the design of ITs. In particular, Ref. [8] focuses on the design of ITs for railway applications. Predictive maintenance to prevent IT faults is studied in [10,11]. Finally, IT calibration and characterization are tackled by authors in [12,13].

This paper is a technical extension of [14], which demonstrated the effectiveness of a new characterization procedure based on a previous author’s idea described in [15]. In summary, the method acquires the frequency response of the IT under test with a single measurement. The frequency range is set by the operator and can be changed depending on the application. Therefore, this procedure saves time and is consequently cost-effective compared to other methods, such as frequency sweep or impulse response, which can be time-consuming and technically complex, respectively. It is well known that the characterization of ITs has been addressed by several researchers [16,17] and is partially described in the relevant standards. Specifically, IEC 61869 is fully dedicated to ITs. General requirements can be found in IEC 61869-1 [18] for ITs and in IEC 61869-6 [19] for the low-power version of ITs (LPITs). The testing procedure used in [14], validated on Rogowski coils but extendable in application to any type of IT, is designed to simplify the existing procedures, reducing the testing time and cost. Such benefits were obtained in [14] without affecting the accuracy of the procedure. The aim of this study is to further validate the applicability of the proposed procedure. This has been accomplished by subjecting the method to a complete uncertainty evaluation, considering the existing uncertainty sources, and propagating their effects on the parameters of interest (ratio error/output voltage and phase displacement). This paper describes in detail the considered uncertainty sources; the Monte Carlo method (MCM) is then used to propagate the uncertainty from the waveform level to the output voltage and phase displacement calculated from the current measurements. This study adds value in two main ways: by providing the reader with a simple procedure to propagate uncertainty that can be replicated for any other kind of ITs and by proving the accuracy of the proposed procedure to confirm its applicability. The final user will then be able to adopt the novel characterization method with the certainty that its results are metrologically valid. This last aspect is quite important for a metrologist, and the concept is described elsewhere in the literature [20,21,22,23].

The remainder of this paper is structured as follows: Section 2 recalls the scenario on which [14] was based and describes the novel motivation of the paper. The characterization procedure is briefly described in Section 3. For more details, the reader may refer to [14]. Section 4 is fully dedicated to the uncertainty evaluation. This section includes the uncertainty sources analysis, the implementation of the MCM, and the discussion of the obtained results. Finally, the conclusion is given in Section 5.

## 2. Scenario

### 2.1. The Rogowski Coil

Rogowski coils are widely used in power systems for AC current measurements. They are known (i) for their flexibility, making it easy to wrap them around conductors of various sizes and shapes, (ii) for their high accuracy in measuring large currents without saturation or heat generation, and finally (iii) for their non-intrusiveness, avoiding physical connections to the conductor. However, the installation and calibration of Rogowski coils can be challenging, requiring technical expertise. They are also sensitive to external magnetic fields, potentially causing measurement errors if not shielded properly. Despite these challenges, Rogowski coils are still a favored choice for current measurement in power systems due to their unique advantages [24,25].

### 2.2. Standards

IEC 61869-10 [26] specifically addresses Rogowski coils for measurement purposes. This standard outlines requirements for their construction, accuracy, temperature range, and electromagnetic compatibility. It provides guidelines for manufacturing and testing procedures, as well as specifications for voltage ratings and insulation coordination. Additionally, Refs. [18,19] include general requirements that also apply to Rogowski coils. These standards help ensure consistency and reliability in the use of Rogowski coils across various power systems, facilitating accurate current measurement and system monitoring. Compliance with these standards is crucial for guaranteeing the interoperability and accuracy of Rogowski coils in power system applications. As far as the uncertainty evaluation is concerned, standards do not provide any indication of the method to implement or prefer. This can be explained by the fact that more than one procedure exists, and the user may select one depending on the available information. In the following, the MCM is adopted according to the suggestions given in the guide to the expression of uncertainty in measurements (GUM) [27] and its Supplement 1 [28].

### 2.3. Motivation

The idea behind this study and its companion paper [14] is to simplify the testing procedure for the characterization of Rogowski coils without neglecting metrological aspects. The characterization in [14] was focused on the harmonic components, yet its applicability to other frequency components is easily achievable without further complications. The advantages of a simplified procedure are many: (i) reduction in the testing time and cost, (ii) increase in the number of users capable of implementing the testing technique, (iii) increase in the number of tested devices, etc. The previous study is complemented here by a comprehensive uncertainty evaluation that confirms the performance of the characterization, enabling its implementation in all applications dealing with ITs.

## 3. Novel Approach

This section recalls the proposed testing approach from [14]. To this end, the measurement setup and the performed tests are presented in dedicated paragraphs.

### 3.1. The Approach

The foundational elements of the proposed approach were developed in [15] (and other previous articles of the authors); a summary is provided here. The concept involves injecting a windowed sinc signal (WSS) that is specifically designed to test the desired frequency range with the desired frequency resolution. In the frequency domain, the spectrum of the sinc function has a rectangular shape, wherein all frequency components outside this rectangular window are zero.

The response of the Rogowski coil to this signal, referred to as the sinc response (SR), is the frequency response of the Rogowski coils under test (RUTs). According to the theory, the SR encapsulates the Rogowski model. Therefore, this model can then be utilized to estimate the Rogowski coil’s response to any input signal within the frequency range defined during the design of the WSS. The estimate of the Rogowski coil’s output can be obtained either in the time domain y(t), employing the convolution operation, or in the frequency domain $Y\left(f\right)$, utilizing the product of the frequency components. In this study, the generic input signal X(f) and the SR of the Rogowski coil SR(f) were convolved as follows:(1)$$Y\left(f\right)=X\left(f\right)\;\times \;SR\left(f\right)$$
where X(f) and SR(f) are the input signal and the sinc response in the frequency domain, respectively. 

In light of the above, the SR was obtained at different operating temperatures in [12]. In particular, −5 °C, 20 °C, and 40 °C were the selected temperatures (according to the temperature classes given in [18]). Consequently, because the SR approach is a single test method with only three tests, it was possible to obtain the Rogowski harmonic model over three different temperatures. The duration of the single test depended on the target frequency resolution. However, being a harmonics a multiple integers of the 50 Hz component, 20 ms is the minimum suitable length for the acquisition window.

### 3.2. Measurement Setup

The measurement setup adopted in this study is described in Figure 1. 

It consists of a function generator Rigol 1022Z with a frequency resolution of 1 µHz; a transconductance Fluke 52120A in metrological confirmation with a maximum current of 120 A; a reference resistive 1 mΩ-shunt with a maximum uncertainty of 2 × 10^−6^ Ω; a National Instruments data acquisition board NI-9238 which has the following main features: (i) 4 input channels; (ii) 24 bits; (iii) full scale of ±0.5 mV; (iv) maximum sampling frequency of 50 kS/s/ch; a climatic chamber with an operating temperature range of −40 °C to 180 °C, and a humidity operating range of 10% to 98%. Finally, the setup includes three commercial RUTs, which characteristics are collected in Table 1.

In a nutshell, the function generator and the transconductance were used to generate the desired current signal. The output of the commercial Rogowski coils and the reference shunt were collected by the acquisition system.

### 3.3. Overview of the Tests

Two sets of tests were performed. One was the implementation of the proposed procedure for testing the devices under test. The other was the reference method.

#### 3.3.1. Proposed Characterization Method

The idea was to measure the SR of the RUTs. The function generator provided a suitable WSS (with the harmonic content of interest), which had a period of 20 ms. Then, the generated voltage signal was amplified and transformed into a current by the transconductance. For each temperature, the WSS was injected, and the output of the RUTs and shunt were measured by the data acquisition board. Each acquisition lasted 1 s. Therefore, 50 SRs were collected to ensure repeatability.

Regarding the test temperatures, the values of −5 °C, 20 °C, and 40 °C were chosen based on the RUT specifications and on the limits given in [18]. Preliminary tests were conducted to estimate the thermal constant of the RUT. The results indicated that a time interval of 3 h was sufficient to achieve thermal stability. The temperature variation rate between any two temperature values was set at 0.1 °C/min.

#### 3.3.2. Reference Method

The chosen reference method was the frequency sweep. This technique requires one measurement for each frequency of interest. Therefore, for each temperature value, measurements up to the number of the many frequencies of interest must be performed. Consequently, the overall duration of the measurements lasts a few minutes in the case of an automatic system or several minutes in the case of a human operator. By contrast, the SR approach requires one measurement that lasts 20 ms for each tested temperature to obtain the desired portion of the frequency response of the IT under test.

For simplicity, only a limited selection of frequencies within the range of 50 Hz to 2500 Hz was tested. The specific frequencies tested are listed in Table 2. 

It should be noted that each test utilized a sinusoidal waveform comprising a single frequency component, with the amplitude specified in the table. The measurement setup employed for the reference test is the same as depicted in Figure 1. However, instead of using a function generator, a calibrator was utilized as the reference instrument. Similar to the approach taken in the proposed procedure, (i) an acquisition window of 1 s was set, and (ii) the outputs of shunt and the RUTs were stored.

#### 3.3.3. Analysis of the Results

The results described in [14] were based on the computation of the typical parameters adopted for ITs, i.e., the ratio error and the phase displacement:(2)$$∆\%=\frac{u_{E}-u_{R}}{u_{R}}*100$$
(3)$$∆\varphi =\varphi_{E}-\varphi_{R}$$

$u_{E}$ and $\varphi_{E}$ are the magnitude and phase estimated by the SR method, while $u_{R}$ and $\varphi_{R}$ are the magnitude and phase obtained from the reference test. For the sake of brevity, Figure 2 and Figure 3 recall a set of results from [14]. The two graphs contain the ∆% and ∆φ of the RUT 1, vs. frequency and vs. temperature, respectively. To facilitate the reader’s comprehension, Table 3 includes some of the limits given in [19] for the accuracy validation vs. harmonic components (the reader may be interested in other documents defining power quality limits as in IEEE Std 519 [29]). From the presented results, two main conclusions were drawn. First, the proposed technique can effectively identify subtle fluctuations in both magnitude and phase. Second, upon closer examination of the frequency, it was noticed that the least desirable outcome occurs at 50 Hz. Note that this frequency component was intentionally included in the designed sinc waveform to enable thorough discussion. However, manufacturers and users frequently need to assess their devices at 50 Hz across different operational scenarios; hence, introducing the 50 Hz component into harmonic testing does not offer any advantages. Therefore, when implementing the suggested approach, it is reasonable to exclude the 50 Hz component from the result evaluation.

## 4. Uncertainty Evaluation

This section is dedicated to quantifying the effectiveness of the proposed characterization procedure in a rigorous manner, considering all the sources of uncertainty. This process is commonly referred to as uncertainty propagation. Among the techniques for evaluating how uncertainty propagates, the chosen method is the MCM, which is described in Supplement 1 [28] of the GUM. In this paper, the MCM is used for analyzing the propagation of the uncertainty along the measurement setup used for both the proposed characterization procedure and the reference test. After the identification of the sources of uncertainty, this approach allows the calculation of confidence intervals associated with the measurement results.

### 4.1. Uncertainty Source Analysis

To identify the sources of uncertainty, it is necessary to refer to Figure 1. In the proposed measurement setup, the data acquisition system (DAQ), and the reference shunt resistor are identified as factors contributing to the uncertainty. To quantitatively assess these contributions, Table 4 lists the accuracy parameters of the DAQ and Table 5 summarizes the performance specifications of transconductance when controlled by a single calibrator as a master unit. 

Finally, Figure 4 and Figure 5 represent the characterization of the reference shunt resistor in the frequency range of interest. Figure 4 depicts amplitude characterization, whereas Figure 5 presents phase-angle characterization. In detail, 25 single-tone sinusoidal currents were applied to the shunt at different frequencies using the transconductance (driven by the calibrator), and the output voltage of the shunt was then measured by the DAQ. Throughout this characterization process, an additional input from the calibrator was acquired as a reference signal for calculating the phase displacement of the shunt resistor. Therefore, the resistance values and the phase displacement of the shunt were defined at various frequencies using the measured voltage and the knowledge of the input current. As shown in Figure 4, the amplitude behavior was stable across frequencies, whereas Figure 5 shows that the phase displacement exhibited some drift across frequencies. In conclusion, thanks to this characterization, it was possible to compute Equation (4), which made it possible to calculate the WSS generated by the transconductance in the frequency domain.

### 4.2. MCM Application

Once the sources of uncertainty had been established, the MCM could be implemented. In brief, the approach involved the corruption of the acquired voltage waveforms during the test by means of the instrumentation’s accuracy parameters (see Table 4 and Table 5). Finally, the output voltage and the phase displacement of the RUTs were calculated in both cases—(i) the proposed method and (ii) the reference method—to assess the accuracy of the estimation of the proposed method compared to the reference method.

Firstly, the procedure consisted of the uncertainty propagation of the value of the shunt resistor in the power quality frequency range. Operationally, this was achieved as follows: (i) the primary current was corrupted by the transconductance accuracy parameters, and (ii) the secondary voltage was corrupted by the DAQ accuracy parameters.

Secondly, uncertainty propagation continued by corrupting (i) the secondary voltage of the shunt by means of the resistor values calculated previously (implemented for both methods) and (ii) the secondary voltage of the RUTs via the DAQ accuracy parameters for both methods. In particular, corruption of the secondary voltage of the shunt was performed as follows:(4)$${\dot{I}}_{p,h}=\frac{{\dot{V}}_{s,h}}{{\dot{Z}}_{h}}$$

In Equation (4), the Fourier transform was applied to the various quantities. ${\dot{V}}_{s,h}$ is the frequency spectrum of secondary voltage measured from the shunt at the frequency having order h. ${\dot{Z}}_{h}$ is the value of the shunt resistor at harmonic order h. ${\dot{I}}_{p,h}$ is the frequency spectrum of the primary current measured thanks to the shunt, which also flows through the RUTs. For the sake of clarity, ${\dot{I}}_{p,h}$ represents the components of the WSS frequency spectrum for the proposed method. Meanwhile, for the reference method, ${\dot{I}}_{p,h}$ represents the components of the sinusoidal waveforms’ frequency spectrum listed in Table 2. In addition, for the latter case, ${\dot{I}}_{p,h}$ is also the generic input signal X(f) of Equation (1).

In conclusion, 20,000 Monte Carlo iterations were performed to calculate the following quantities from the RUTs: (i) the output voltage $V_{rog}$, (ii) the phase displacement ${∆\varphi }_{rog}$, and (iii) their 95% confidence intervals.

### 4.3. MCM Results and Discussion

In this paper, the MCM was used to evaluate the reliability of the estimate obtained from the proposed method with respect to the reference method. The outputs of all MCM results are listed in Table 6 and Table 7 for RUT 1 (for the sake of completeness and repeatability). For each temperature and frequency, these tables include the estimated (Est.) and reference (Ref.) values in terms of the mean and the standard deviation (std) derived from the 20,000 Monte Carlo iterations. In addition, “Max.” and “Min.” values delineate the upper and lower bounds of the shortest interval containing 95% of the MC results. In particular, Table 6 contains the output voltage $V_{rog}$ of the RUT 1, while Table 7 contains the phase displacement ${∆\varphi }_{rog}$ of the RUT 1. Some of the results are better visualized in a later figure. 

The reported results are expressed using the number of significant digits correlating to their associated standard deviation. Considering the amplitudes in Table 6, the values obtained from the proposed method match those from the reference test for all temperatures and frequencies; in fact, the confidence intervals overlap. This guarantees the effectiveness of the method regarding the estimated output voltage $V_{rog}$ of RUT 1. For the sake of brevity, the results of the other RUTs are not reported, but the same considerations apply. In conclusion, the sinc-response method is valid, accurate, and applicable for evaluating the output voltage of Rogowski coils. The confidence intervals for the phase displacement ${∆\varphi }_{rog}$ of the RUT 1 do not quite overlap with those of signals ranging up to 350 Hz. Our initial observation here is that the proposed method demonstrates its validity from 550 Hz to 2500 Hz for all temperatures, while in the 50 Hz–350 Hz range, the effectiveness depends on the temperature. Therefore, our second observation focuses on the link between temperature and the sinc-response method. Specifically, the proposed method appears to be less effective at high temperatures than at low temperatures. This behavior is clearly depicted in Figure 6, where blue dots represent phase displacement estimates, while the red dots represent the results of the reference case. For space reasons, only two frequencies (i.e., 150 and 250 Hz) are reported at three different temperatures (i.e., 20 °C, 40 °C, and −5 °C), but these frequencies are the most significant, as they represent the worst-case scenario. In Figure 6, the error bars are not clearly visible, since the confidence intervals are quite narrow, as evident in Table 7. The results for the output voltage of the other RUTs are likewise not reported, but the same considerations apply.

The issue with the 50 Hz value was already discussed in [12] and in the previous section. In fact, the method was proposed for the evaluation of harmonic components, and the 50 Hz signal was included for the sake of discussion. Two observations can be made regarding the out-of-spec phase angle values: First, the instrumentation adopted is quite accurate, and hence, the resulting confidence intervals are extremely narrow (consider that in the 95% confidence interval, the two intervals overlap). The second observation concerns the significance of the phase angle of the harmonic components. As Table 3 indicates, there is considerable flexibility for estimating the phase angle of the harmonic components. Furthermore, the estimated values obtained from the method are far below those limits.

Overall, the proposed method is highly effective at estimating the amplitude and phase of the harmonic components tested.

## 5. Conclusions

This paper, serving as a technical extension of the study presented in [14], focuses on the evaluation of uncertainty in a novel characterization method. The method introduces frequency-based characterization through the unique approach suggested by the authors, specifically tailored for testing instrument transformers. The performance of this novel procedure was examined using three commercially available Rogowski coils, each at three distinct temperatures conforming to established standards. In this paper, the method was validated from a metrological perspective and subjected to a comprehensive uncertainty assessment to gauge its accuracy and feasibility. To achieve this, the Monte Carlo method was applied in accordance with the Guide to the Expression of Uncertainty in Measurement and its Supplement 1. The results reveal that the proposed frequency characterization method is not only effective but also highly accurate, demonstrating its suitability for widespread implementation across various applications involving instrument transformers. While minor limitations were found in the phase angle estimation, these did not undermine the performance of the method. It is essential to note that this uncertainty analysis will play a crucial role when considering the method for potential inclusion in future revisions of industry standards.

## Figures and Tables

**Figure 1 sensors-24-01746-f001:**
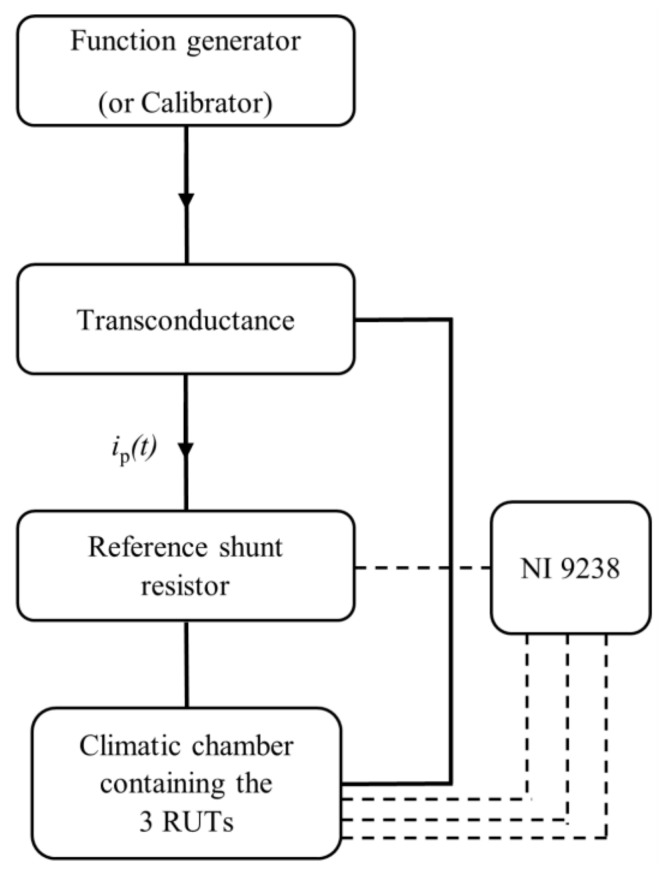
Schematic of the measurement setup.

**Figure 2 sensors-24-01746-f002:**
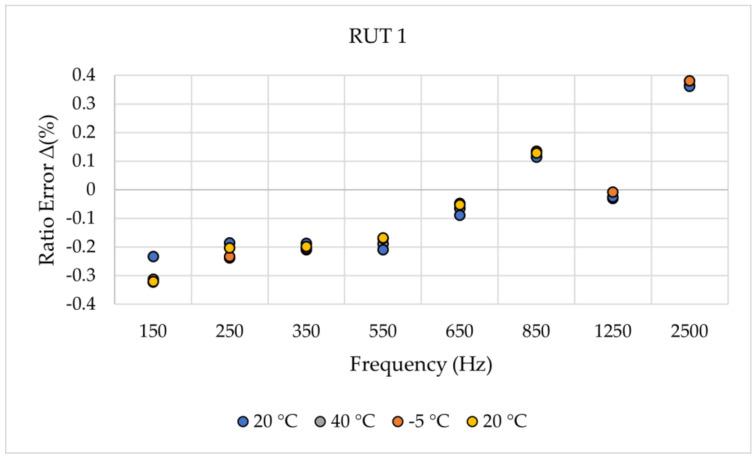
Ratio error ∆% of RUT 1 for different frequencies and temperatures [14].

**Figure 3 sensors-24-01746-f003:**
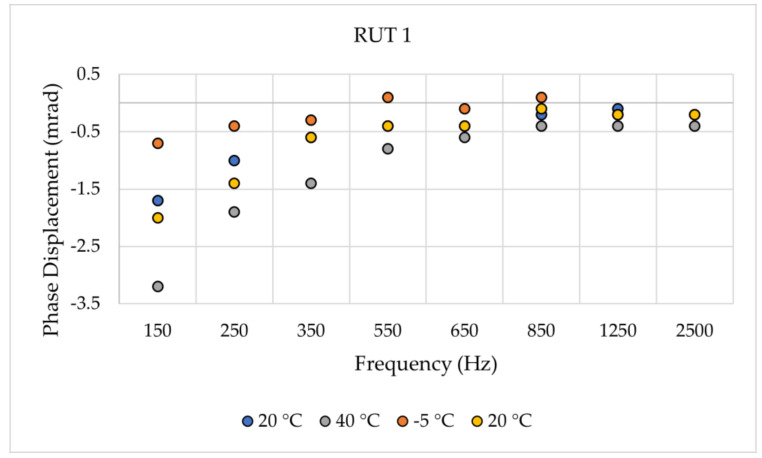
Phase displacement ∆φ of RUT 1 for different frequencies and temperatures [14].

**Figure 4 sensors-24-01746-f004:**
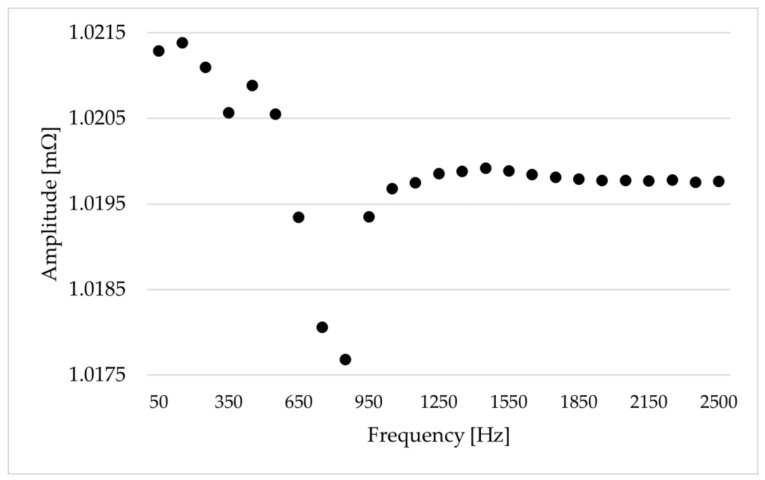
Resistance values vs. frequency of the shunt resistor.

**Figure 5 sensors-24-01746-f005:**
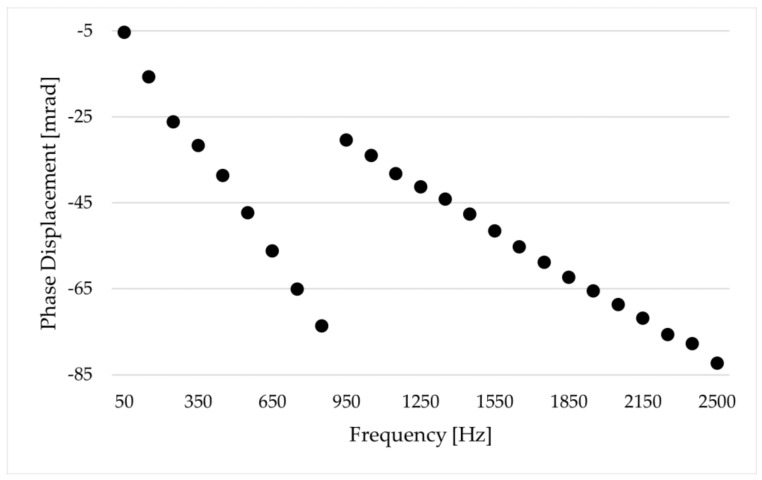
Phase displacement vs. frequency of the shunt resistor.

**Figure 6 sensors-24-01746-f006:**
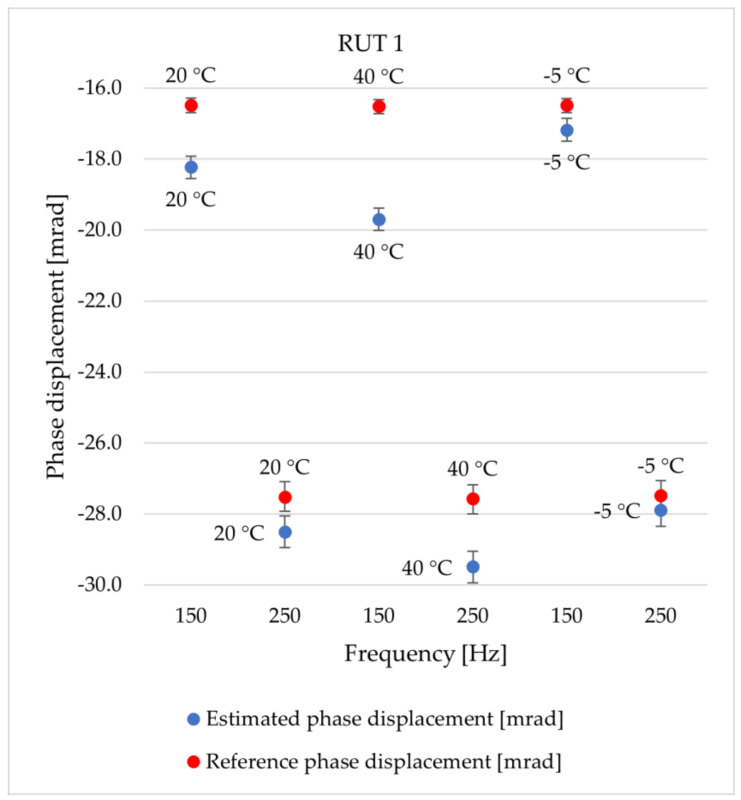
${∆\varphi }_{rog}$ at 150 Hz and 250 Hz and at tested temperatures for RUT 1.

**Table 1 sensors-24-01746-t001:** Main features of the RUTs.

	RUT 1	RUT 2	RUT 3
Accuracy [%]	±0.5	±1	±1
Output voltage at 50 Hz [mV/kA]	85	100	100
Type	Flexible split core	Flexible split core	Flexible split core
Rated current [A]	1000	–	1000
Inner diameter [mm]	100	120	68

**Table 2 sensors-24-01746-t002:** Characteristics of the signals used for the reference tests.

Frequency [Hz]	Amplitude [A]
50	100
150	50
250	50
350	50
550	50
650	50
850	50
1250	36
2500	36

**Table 3 sensors-24-01746-t003:** Ratio error and phase error limits given in [19].

Accuracy Class	Ratio Error at Harmonic [%]				
	2nd to 4th	5th and 6th	7th to 9th	10th to 13th	Above 13th
0.5	5	10	20	20	+20 −100
1	10	20	20	20	+20 −100
	Phase displacement at harmonic [mrad]				
0.5	87	174	349	349	–
1	174	349	349	349	–

**Table 4 sensors-24-01746-t004:** Performance specifications of the DAQ.

Gain Error [%]	Offset Error [%]	Noise [µV]
0.07	0.005	3.9

**Table 5 sensors-24-01746-t005:** Performance specifications of transconductance.

Frequency [Hz]	Output [%]	Range [%]	Phase Angle Accuracy [°]	Distortion [mA]	Noise [dB]
10 to 69	0.011	0.003	0.006	2.5	−70
69 to 180			0.012		
180 to 450			0.025		
450 to 850			0.045		
850 to 6000	0.052	0.005	0.325	39.7	

**Table 6 sensors-24-01746-t006:** $V_{rog}$ for every temperature, frequency, and signal for RUT 1.

Temp. [°C]	Freq. [Hz]	Signal	Min. [mV]	Mean [mV]	Max. [mV]	Std [mV]
20	50	Est.	7.095	7.102	7.109	0.004
		Ref.	7.096	7.100	7.105	0.003
20	150	Est.	10.650	10.658	10.666	0.004
		Ref.	10.645	10.652	10.659	0.004
20	250	Est.	17.747	17.759	17.771	0.007
		Ref.	17.743	17.754	17.766	0.007
20	350	Est.	24.84	24.86	24.88	0.01
		Ref.	24.84	24.86	24.88	0.01
20	550	Est.	39.04	39.07	39.10	0.02
		Ref.	39.04	39.07	39.09	0.02
20	650	Est.	46.15	46.18	46.21	0.02
		Ref.	46.14	46.17	46.20	0.02
20	850	Est.	60.36	60.40	60.44	0.02
		Ref.	60.34	60.38	60.42	0.02
20	1250	Est.	63.91	63.95	63.99	0.03
		Ref.	63.89	63.94	63.98	0.03
20	2500	Est.	127.85	127.93	128.02	0.05
		Ref.	127.82	127.90	127.99	0.05
40	50	Est.	7.105	7.112	7.120	0.004
		Ref.	7.101	7.106	7.111	0.003
40	150	Est.	10.653	10.660	10.668	0.004
		Ref.	10.653	10.660	10.667	0.004
40	250	Est.	17.762	17.774	17.787	0.007
		Ref.	17.757	17.768	17.781	0.007
40	350	Est.	24.86	24.88	24.89	0.01
		Ref.	24.86	24.88	24.89	0.01
40	550	Est.	39.08	39.10	39.13	0.02
		Ref.	39.07	39.10	39.12	0.02
40	650	Est.	46.19	46.22	46.25	0.02
		Ref.	46.18	46.21	46.24	0.02
40	850	Est.	60.39	60.43	60.47	0.02
		Ref.	60.39	60.43	60.47	0.02
40	1250	Est.	63.96	64.00	64.04	0.03
		Ref.	63.95	63.99	64.03	0.03
40	2500	Est.	127.93	128.02	128.10	0.05
		Ref.	127.92	128.01	128.09	0.05
−5	50	Est.	7.077	7.085	7.092	0.004
		Ref.	7.090	7.094	7.099	0.003
−5	150	Est.	10.640	10.648	10.655	0.004
		Ref.	10.636	10.643	10.650	0.004
−5	250	Est.	17.734	17.746	17.758	0.007
		Ref.	17.728	17.740	17.752	0.007
−5	350	Est.	24.83	24.84	24.86	0.01
		Ref.	24.82	24.84	24.86	0.01
−5	550	Est.	39.02	39.04	39.07	0.02
		Ref.	39.01	39.03	39.06	0.02
−5	650	Est.	46.12	46.15	46.18	0.02
		Ref.	46.10	46.13	46.17	0.02
−5	850	Est.	60.31	60.35	60.39	0.02
		Ref.	60.29	60.33	60.37	0.02
−5	1250	Est.	63.86	63.91	63.95	0.03
		Ref.	63.84	63.89	63.93	0.03
−5	2500	Est.	127.77	127.86	127.94	0.05
		Ref.	127.72	127.81	127.89	0.05
20 °C	50	Est.	7.096	7.103	7.110	0.004
		Ref.	7.095	7.100	7.105	0.003
20 °C	150	Est.	10.641	10.648	10.656	0.004
		Ref.	10.644	10.652	10.659	0.004
20 °C	250	Est.	17.741	17.753	17.765	0.007
		Ref.	17.743	17.754	17.766	0.007
20 °C	350	Est.	24.85	24.86	24.88	0.01
		Ref.	24.84	24.86	24.87	0.01
20 °C	550	Est.	39.05	39.08	39.10	0.02
		Ref.	39.04	39.07	39.09	0.02
20 °C	650	Est.	46.14	46.18	46.21	0.02
		Ref.	46.14	46.17	46.20	0.02
20 °C	850	Est.	60.35	60.39	60.43	0.02
		Ref.	60.34	60.38	60.42	0.02
20 °C	1250	Est.	63.90	63.95	63.99	0.03
		Ref.	63.89	63.94	63.98	0.03
20 °C	2500	Est.	127.85	127.94	128.02	0.05
		Ref.	127.82	127.91	127.99	0.05

**Table 7 sensors-24-01746-t007:** ${∆\varphi }_{rog}$ for every temperature, frequency, and signal for RUT 1.

Temp. [°C]	Freq. [Hz]	Signal	Min [mrad]	Mean [mrad]	Max [mrad]	Std [mrad]
20	50	Est.	−13.1	−12.4	−11.8	0.3
		Ref.	−5.6	−5.5	−5.4	0.1
20	150	Est.	−18.5	−18.2	−17.9	0.2
		Ref.	−16.7	−16.5	−16.3	0.1
20	250	Est.	−28.9	−28.5	−28.1	0.3
		Ref.	−27.9	−27.5	−27.1	0.3
20	350	Est.	−34.6	−34.2	−33.8	0.3
		Ref.	−34.0	−33.6	−33.2	0.3
20	550	Est.	−51.7	−50.9	−50.2	0.5
		Ref.	−51.3	−50.5	−49.8	0.5
20	650	Est.	−61.2	−60.5	−59.7	0.5
		Ref.	−60.8	−60.0	−59.3	0.5
20	850	Est.	−79.6	−78.9	−78.1	0.5
		Ref.	−79.4	−78.7	−77.9	0.5
20	1250	Est.	−54	−49	−44	3
		Ref.	−54	−49	−44	3
20	2500	Est.	−103	−98	−92	3
		Ref.	−103	−98	−92	3
40	50	Est.	−16.7	−16.0	−15.4	0.3
		Ref.	−5.6	−5.5	−5.4	0.1
40	150	Est.	−20.0	−19.7	−19.4	0.2
		Ref.	−16.7	−16.5	−16.3	0.1
40	250	Est.	−29.9	−29.5	−29.1	0.3
		Ref.	−28.0	−27.6	−27.2	0.3
40	350	Est.	−35.5	−35.1	−34.6	0.3
		Ref.	−34.0	−33.6	−33.2	0.3
40	550	Est.	−52.1	−51.4	−50.7	0.5
		Ref.	−51.3	−50.6	−49.8	0.5
40	650	Est.	−61.4	−60.7	−60.0	0.5
		Ref.	−60.8	−60.1	−59.3	0.5
40	850	Est.	−79.9	−79.2	−78.4	0.5
		Ref.	−79.5	−78.8	−78.0	0.5
40	1250	Est.	−55	−49	−44	3
		Ref.	−55	−49	−44	3
40	2500	Est.	−103	−98	−93	3
		Ref.	−103	−98	−92	3
−5	50	Est.	−9.3	−8.6	−7.9	0.3
		Ref.	−5.6	−5.5	−5.4	0.1
−5	150	Est.	−17.5	−17.2	−16.9	0.2
		Ref.	−16.7	−16.5	−16.3	0.1
−5	250	Est.	−28.3	−27.9	−27.4	0.3
		Ref.	−27.9	−27.5	−27.1	0.3
−5	350	Est.	−34.2	−33.8	−33.4	0.3
		Ref.	−33.9	−33.5	−33.1	0.3
−5	550	Est.	−51.1	−50.4	−49.6	0.5
		Ref.	−51.2	−50.4	−49.7	0.5
−5	650	Est.	−60.8	−60.1	−59.3	0.5
		Ref.	−60.7	−60.0	−59.2	0.5
−5	850	Est.	−79.4	−78.6	−77.9	0.5
		Ref.	−79.4	−78.6	−77.9	0.5
−5	1250	Est.	−54	−49	−43	3
		Ref.	−54	−49	−43	3
−5	2500	Est.	−103	−98	−93	3
		Ref.	−103	−97	−92	3
20 °C	50	Est.	−13.1	−12.5	−11.8	0.3
		Ref.	−5.6	−5.5	−5.4	0.1
20 °C	150	Est.	−18.8	−18.5	−18.2	0.2
		Ref.	−16.7	−16.5	−16.3	0.1
20 °C	250	Est.	−29.3	−28.9	−28.4	0.3
		Ref.	−27.9	−27.5	−27.1	0.3
20 °C	350	Est.	−34.6	−34.2	−33.8	0.3
		Ref.	−34.0	−33.6	−33.2	0.3
20 °C	550	Est.	−51.7	−51.0	−50.2	0.5
		Ref.	−51.3	−50.5	−49.8	0.5
20 °C	650	Est.	−61.1	−60.4	−59.7	0.5
		Ref.	−60.8	−60.0	−59.3	0.5
20 °C	850	Est.	−79.6	−78.9	−78.1	0.5
		Ref.	−79.5	−78.7	−78.0	0.5
20 °C	1250	Est.	−54	−49	−44	3
		Ref.	−54	−49	−43	3
20 °C	2500	Est.	−103	−98	−92	3
		Ref.	−103	−98	−92	3

## Data Availability

Data contained within the article.

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
