# Peer review of "Validation of a Simplified Method for Estimating the Harmonic Response of Rogowski Coils with the Monte Carlo Method"

_sensors, 2024, doi:10.3390/s24061746_

Round 1
Reviewer 1 Report
Comments and Suggestions for Authors
This paper introduces a novel frequency-based characterization method for testing Instrument Transformers and provides a comprehensive uncertainty assessment using Monte Carlo methods. The engineering content of this article is rich in data; however, minor adjustments in writing style are recommended before publication. Below are the specific suggestions:
1. In lines 130-132, the description employs time-domain symbols, while Equation (1) is in the frequency domain. It is advised to maintain consistency in terminology.
2. Punctuation should not be added after numbered equations.
3. The red line beneath iP(t) in Fig.1 should be removed.
4. Fig.2 and Fig.3 are referenced from cited literature; it is advisable to indicate the referenced literature in the figure titles.
Comments on the Quality of English LanguageMinor editing of English language required.
Author Response
- In lines 130-132, the description employs time-domain symbols, while Equation (1) is in the frequency domain. It is advised to maintain consistency in terminology.
- Punctuation should not be added after numbered equations.
- The red line beneath iP(t) in Fig.1 should be removed.
- 2 and Fig.3 are referenced from cited literature; it is advisable to indicate the referenced literature in the figure titles.
Thank you for the effort and time spent reviewing the paper. The suggestions were accepted and implemented.
Reviewer 2 Report
Comments and Suggestions for Authors
Please find the comments referring to the paper as an attachment.

Author Response
- Abstract section or Introduction section – please more emphasize the main purpose of the research in terms of the theoretical/practical applications.
- Introduction section – please replace the notation [x‐y] by the notation [x–y], e.g. line 71. Please check the entire article for this issue.
- Please mark the minus sign with the symbol: – , e.g. line 168. Please check the entire article for this issue.
- Please mark the product symbol with a dot instead an asterisk, e.g. Eq. (2) and line 147. Please check the entire article for this issue.
- Figure 1 – please replace the notation i p(t) by the notation ip(t). Physical quantities should be written in italics. Please check the entire article for this issue.
- Descriptions in Figures should have a font size at most equal to the font size in the body of the article, e.g. Figures 2 and 3. Please check the entire article for this issue.
- Data included in Tables should have a font size at most equal to the font size in the body of the article, e.g. Table 2. Please check the entire article for this issue.
- If the subscripts are not a physical variable but an abbreviation of a proper name, please do not write these symbols in italics, e.g. lines: 277 and 278. Please check the entire article for this issue.
Thank you for the effort and time spent reviewing the paper. We implemented all the comments and suggestions in the new version of the paper.
- Tables 6 and 7 – what is the reason for the different number of significant digits for the results included in these Tables?
The standard deviation is an estimate of the standard uncertainty associated with the output quantity of the Monte Carlo Method. Therefore, Tables 6 and 7 are reported with a number of significant digits coherent with the standard deviation. This is a trivial (mandatory) practice whenever metrological aspects are involved.
- Line 261 – what DAQ accuracy parameters are recommended to obtain reliable calculation results?.
The standards do not provide minimum specifications regarding the data acquisition system. In any case, the chosen DAQ allows to measure voltages with a maximum uncertainty negligible compared to the accuracy classes of the adopted devices.
- Line 286 – what is the justification for the assumed iterations for the Monte Carlo method?. Is this the number of iterations consistent with the guidelines in the guide [28].
The supplement of GUM [28] suggests that the number of Monte Carlo iterations should be equal or greater than 1/(1-p) where p is the probability. As for our application, preliminary tests confirmed that a larger number of trials do not result in better performance.
Reviewer 3 Report
Comments and Suggestions for Authors
The uncertainty analysis of a novel characterization process for instrument transformer testing is the main topic of this paper. Three commercially available Rogowski coils were tested at three distinct temperatures. The accuracy and viability of the approach were confirmed through metrological validation.
The manuscript is well organized, and significant works are cited in the context of the current state of research in the field.
The figures are clear, intuitive, easy to interpret.
The work fulfills the stated objective.
It is recommended to be accepted.
Author Response
The uncertainty analysis of a novel characterization process for instrument transformer testing is the main topic of this paper. Three commercially available Rogowski coils were tested at three distinct temperatures. The accuracy and viability of the approach were confirmed through metrological validation.
The manuscript is well organized, and significant works are cited in the context of the current state of research in the field.
The figures are clear, intuitive, easy to interpret.
The work fulfills the stated objective.
It is recommended to be accepted.
We thank the reviewer for the nice comments and for the time spent reviewing the paper.
Round 2
Reviewer 2 Report
Comments and Suggestions for Authors
All comments of the reviewer have been included in the revised version of the paper. I recommend publication of this paper in its current form.